# COVID-Net L2C-ULTRA: An Explainable Linear-Convex Ultrasound Augmentation Learning Framework to Improve COVID-19 Assessment and Monitoring

**DOI:** 10.3390/s24051664

**Published:** 2024-03-04

**Authors:** E. Zhixuan Zeng, Ashkan Ebadi, Adrian Florea, Alexander Wong

**Affiliations:** 1Department of Systems Design Engineering, University of Waterloo, Waterloo, ON N2L 3G1, Canada; alexander.wong@uwaterloo.ca; 2Digital Technologies Research Centre, National Research Council Canada, Toronto, ON M5T 3J1, Canada; 3Department of Emergency Medicine, McGill University, Montreal, QC H4A 3J1, Canada; adrian.florea@mail.mcgill.ca; 4Waterloo Artificial Intelligence Institute, Waterloo, ON N2L 3G1, Canada

**Keywords:** lung ultrasonic imaging, linear–convex augmentation, COVID-19 assessment, deep explainable architecture

## Abstract

While no longer a public health emergency of international concern, COVID-19 remains an established and ongoing global health threat. As the global population continues to face significant negative impacts of the pandemic, there has been an increased usage of point-of-care ultrasound (POCUS) imaging as a low-cost, portable, and effective modality of choice in the COVID-19 clinical workflow. A major barrier to the widespread adoption of POCUS in the COVID-19 clinical workflow is the scarcity of expert clinicians who can interpret POCUS examinations, leading to considerable interest in artificial intelligence-driven clinical decision support systems to tackle this challenge. A major challenge to building deep neural networks for COVID-19 screening using POCUS is the heterogeneity in the types of probes used to capture ultrasound images (e.g., convex vs. linear probes), which can lead to very different visual appearances. In this study, we propose an analytic framework for COVID-19 assessment able to consume ultrasound images captured by linear and convex probes. We analyze the impact of leveraging extended linear-convex ultrasound augmentation learning on producing enhanced deep neural networks for COVID-19 assessment, where we conduct data augmentation on convex probe data alongside linear probe data that have been transformed to better resemble convex probe data. The proposed explainable framework, called COVID-Net L2C-ULTRA, employs an efficient deep columnar anti-aliased convolutional neural network designed via a machine-driven design exploration strategy. Our experimental results confirm that the proposed extended linear–convex ultrasound augmentation learning significantly increases performance, with a gain of 3.9% in test accuracy and 3.2% in AUC, 10.9% in recall, and 4.4% in precision. The proposed method also demonstrates a much more effective utilization of linear probe images through a 5.1% performance improvement in recall when such images are added to the training dataset, while all other methods show a decrease in recall when trained on the combined linear–convex dataset. We further verify the validity of the model by assessing what the network considers to be the critical regions of an image with our contribution clinician.

## 1. Introduction

While no longer a public health emergency, coronavirus disease 2019 (COVID-19) remains an established and ongoing global health threat. The COVID-19 pandemic, especially with the recent rise of its variants, e.g., Omicron, has resulted in significant negative impacts on the health and well-being of the global population. While PCR and rapid antigen tests have played crucial roles in COVID-19 screening [1], a major aspect of the COVID-19 clinical workflow is imaging-based assessment. The benefits of imaging-based assessment include the ability to provide visual evidence of lung abnormalities associated with COVID-19, aiding in assessing disease severity and progression. Additionally, imaging can be particularly useful in cases where PCR and antigen tests yield negative results or for individuals with severe symptoms or complications [2]. There has been a significant increase in the use of point-of-care ultrasound (POCUS) imaging alongside chest radiography due to its low cost, low maintenance, disinfection ease, portability, and lack of ionizing radiation [3]. However, a major barrier to the widespread adoption of POCUS in the COVID-19 clinical workflow is the scarcity of expert clinicians who can interpret POCUS examinations for COVID-19 assessment [4]. Therefore, there has been considerable interest in deep learning (DL)-driven clinical decision support systems [5].

DL-driven clinical decision support systems have shown great potential in addressing the challenge of scarcity of expert clinicians and helping healthcare service providers by providing them with artificial intelligence (AI)-powered tools, augmenting/automating the interpretation of POCUS images (e.g., [5,6]). Convolutional neural networks (CNNs) have been particularly effective in various medical image analysis tasks. Nonetheless, designing CNNs for POCUS-based screening poses unique challenges, one of which is the heterogeneity in the types of probes used to capture ultrasound images. For example, POCUS images captured using convex probes have very different visual appearances than images captured using linear probes. Limiting data to a single probe type increases consistency at the cost of significantly reduced dataset size which could result in reduced accuracy. In addition, using data from different probe types can enable novel insights that might not be possible with just one type of data.

Motivated to tackle this challenge, in this paper, we propose an explainable framework leveraging extended linear–convex ultrasound augmentation learning, called COVID-Net L2C-ULTRA, to produce enhanced deep neural networks for COVID-19 assessment and monitoring. More specifically, we leverage projective and piecewise affine transformations as part of the data augmentation pipeline to transform linear probe data to better resemble convex data, as well as increase the diversity of viewing windows to improve generalization. To evaluate the effectiveness of our proposed method, we designed an efficient deep columnar anti-aliased convolutional neural network via a machine-driven design exploration strategy. Our experiments demonstrate that the extended linear–convex ultrasound augmentation learning technique significantly increases the performance of the neural network, resulting in a gain of 3.9% in test accuracy and 3.2% in the area under the receiver operating characteristic curve (AUC). To guarantee the reliability and responsibility of the network, an explanatory module was developed for evaluating network decisions using visual explanation tools. Furthermore, our collaborating clinician (A.F.) meticulously reviewed and authenticated the pipeline and outcomes to validate the credibility of the proposed solution from a clinical standpoint.

The remainder of this paper is organized as follows: Section 2 provides a brief overview of related work in POCUS-based COVID-19 screening and deep learning-driven clinical decision support systems, including data augmentation and different ultrasound probes. Section 3 describes data used in this study and Section 4 details the methodology of our proposed extended linear–convex ultrasound augmentation learning technique and the architecture of COVID-Net L2C-ULTRA. Section 5 presents the experimental results and performance comparisons, along with clinical validation of model results using explainable AI (XAI) methods. Section 6 concludes the paper by discussing the findings along with limitations and future research directions.

## 2. Background

### 2.1. Data Augmentation

The application of deep learning techniques in computer vision has demonstrated reliable performance in tasks such as classification, object detection, segmentation, and others [7]. However, such methods rely heavily on large, diverse datasets. The MNIST (Modified National Institute of Standards and Technology) handwritten digit database [8], for example, has close to 7000 images for each digit. Imagenet [9], a common general-purpose image benchmark database, contains 14,197,122 images. In contrast, according to the review published in 2018 [4], ultrasound image datasets are usually smaller than 300 for each case and images were taken from one static place with one type of ultrasound device. The high level of specialized technical knowledge for labeling medical data along with patient privacy concerns makes it nearly impossible for a medical dataset to reach the same scale as other image dataset types [10,11]. To combat this issue, data augmentation is a common technique that holds the potential to significantly enhance the performance of deep learning models [12].

Most image data augmentation techniques are created with optical image sensors in mind and may not reflect real changes that may happen in ultrasound. For example, brightness adjustment simulates changes to a scene’s illumination. In an ultrasound image, however, the brightness value is a direct reflection of the tissue condition itself rather than external factors present. A horizontal flip of an image will be the equivalent of rotating the probe 180°, but there is a set convention as to how to orientate probes in various locations. Generally, there is an indicator marker located on one side of the probe, which would be oriented towards the patient’s right side or head [13], and this convention would be disrupted through horizontal flips. Vertical flips would change the origin of the ultrasound rays from the top of the image to the bottom. Tissues at a lower depth would show as being on top. Acoustic shadows and enhancements which lay under particular tissues would then also have reversed directions. Another common image transformation used for data augmentation is rotation, which suffers from the same issue of changing the location of the source of the ultrasound waves. Therefore, the clinical relevance of the common data augmentation techniques when applied to ultrasound images should be carefully considered.

One alternative to those image transformation techniques to augment ultrasound images is by adding and removing naturally occurring artifacts that may appear in ultrasound. Sezer et al. [14] showed success with using optimized Bayesian non-local means (OBNLM) [15] to reduce speckle noise as a data augmentation technique by creating one other set of denoised data. Tirindelli et al. [16] went the opposite route and looked at adding realistic artifacts and noise-reflecting deformations, reverberations, and signal-to-noise-ratio to linear ultrasound images, but did not show significant improvement in image classification. SpeckleGAN [17] also looks to add speckle noise artifacts as a data augmentation technique. Reverberation artifact methods, such as those described in [18,19,20], may also potentially be used for data augmentation.

One disadvantage of such methods is the need to directly modify the image content itself. If not carried out carefully, important image features might be unintentionally removed or altered. More complex methods using generative adversarial networks (GAN) [21] and other deep learning models to identify, add, and/or remove artifacts are also very computationally intensive due to needing a large deep learning network to generate, and can generally only be used to generate another set of data rather than randomly modifying images during training itself. This implies that these models are limited in the amount of diversity they can generate.

### 2.2. Ultrasound Probes

Ultrasound imaging is a versatile and non-invasive modality that has been widely adopted for various clinical applications [22,23,24], including COVID-19 assessment [6,25,26,27]. However, the heterogeneity in the types of ultrasound probes used to capture images can lead to significant differences in the visual appearance and quality of the images [28,29]. Commonly used ultrasound probe types include linear, convex (curvilinear), and phased-array probes.

Linear probes have a rectangular shape, with the transducer elements arranged in a straight line. Linear probes have a high frequency and typically produce a high-resolution image with a narrow field of view, and result in a rectangular image. In contrast, convex probes have a curved or convex shape, and the transducer elements are arranged in an arc. Convex probes have a lower frequency, produce a wider field of view with a lower resolution, and generally have greater penetration depth [28,29]. Due to the difference in transducer element arrangement, results from convex probes have artifacts arranged in a “tilted” fashion, radiating from the focal point of the probe, which results in visual differences of artifacts between the two probe results [30].

Both transducer types serve as a good point-of-care solution for diagnosing chest diseases [31,32], but the difference in visual appearance due to the transducer element arrangement can make it difficult for deep learning models to generalize between them. Prior methods have experimented with “rectifying” convex ultrasound images into rectangular viewing windows similar to linear images [30]. However, given that the majority of images in the public ultrasound datasets (e.g., [33,34]) are generated with convex probes rather than linear, it may be more advantageous to transform the minority probe type instead. This would make it less computationally intensive during inference as well, reducing the need to transform the image for the majority of cases.

## 3. Data

The COVIDx-US v1.4 dataset [33] was used as the data source. COVIDx-US is a public benchmark dataset of lung ultrasound imaging data. The dataset consists of 16,649 processed POCUS images, with 12,260 captured using convex probes and 4389 captured by linear probes. In terms of disease breakdown, 6764 of the POCUS images were COVID-19-positive, and 9985 were COVID-19-negative. These images were gathered from a total of 65 COVID-19-positive and 82 COVID-19-negative unique patients. The train, validation, and testing splits are 72%, 14%, and 14%, respectively. The COVIDx-US dataset also provides a unified and standardized human “gold standard” lung ultrasound score (LUSS). Each split is sampled to ensure a similar distribution of LUSS scores and specific disease assessment (COVID-19 infection, normal, non-COVID-19 infection, other lung diseases/conditions). Each source video for the images, as well as each individual patient, is assigned to only a single split.

## 4. Methods

In this study, we propose an extended linear–convex ultrasound augmentation learning approach, called COVID-Net L2C-ULTRA, to address data scarcity challenges and account for heterogeneity in POCUS images arising from the use of different types of probes. Our method leverages random projective and piecewise affine augmentations to increase the diversity of viewing windows and transform linear probe data to better resemble the visual appearance of convex probe data. This facilitates improved generalization of a deep neural network, as it is exposed to a greater diversity of data in a more visually consistent form.

Extended linear–convex ultrasound augmentation learning aims to utilize the maximum amount of POCUS data while accounting for the visual appearance differences caused by employing different probes for capturing POCUS images. For instance, convex probe images have a larger field of view, with a cone-shaped viewing window that varies in angle and form across devices (Figure 1). Linear probe images, on the other hand, have a more restricted field of view and a linear viewing window [35], leading to distorted visual content compared to convex probe images.

### 4.1. Projective Transform

To deal with these various challenges, we leverage random projective augmentations to not only increase the diversity of viewing windows but also to transform linear probe data in such a way that they better resemble the visual appearance of convex probe data. This approach will enable augmented learning using both convex probe and linear probe data, fostering improved generalization since the deep neural network will experience a wider range of data diversity while maintaining visual consistency. Yaron [30] applies a similar idea by rectifying convex images to appear more similar to linear ones using their polar coordinates. However, the majority of the COVIDx-US dataset (and other public datasets) contains convex images (73.6%). Converting all such images into linear form would introduce more visual distortion in the overall dataset as the top, thinner part of the viewing window is stretched in a greater number of images. In addition, the dataset composition also suggests that convex probes are a more popular choice for looking at chest ultrasounds. In this case, applying a transformation only on linear probes in inference would place a lower computational requirement in the majority of use cases. As such, applying the transformation in inference on only linear probes allows us to reduce the number of images that needs to be distorted.

Furthermore, during training, we can transform the POCUS images obtained via both linear and convex probes using random projective transforms as a data augmentation. This process enhances their visual resemblance to POCUS images captured by a convex probe, allowing the deep neural network to more effectively harness this additional data source. An example result of this POCUS augmentation is shown in Figure 2b,e, with Figure 2a,d as the original images respectively. In comparison to the original, Figure 2b can be seen to have slightly increased the slant of the viewing window, while Figure 2a was transformed into a more conical shape, which is more similar to the appearance of convex probe results.

If the bounds of the viewing window are known, then the POCUS image can be transformed to have various different angles by remapping the corners of the viewing window. To obtain these labels, the boundaries are first approximately labeled manually. Those labels are then processed to ensure that the absolute slope of the left and right boundaries are the same and that the left and right points lie on the same horizontal axis. This symmetry makes the image transformations significantly easier. This point remapping process is shown in Figure 3. Here, new points for each corner of the viewing window are randomly selected, and a transformation matrix is defined accordingly. Specifically, the corners of the original viewing window can be used to find both a center and a slope. A new slope is then randomly sampled based on the original slope. While keeping the center point constant, the new slope is then used to find the corners of the viewing window.

A projective transform is defined by
$$x^{\prime }=Hx$$
where
$$x=\left[\begin{array}{c} x\\ y\\ 1 \end{array}\right],H=\left[\begin{array}{ccc} a_{0} & a_{1} & a_{2}\\ b_{0} & b_{1} & b_{2}\\ c_{0} & c_{1} & 1 \end{array}\right]$$

*H* can be estimated by the four pairs of points from Figure 3 and the following equations:$$0=a_{0}*x+a_{1}*y+a2-c_{0}*x*x^{\prime }-c_{1}*y*x^{\prime }-x^{\prime }$$
$$0=b_{0}*x+b_{1}*y+b2-c_{0}*x*y^{\prime }-c_{1}*y*y^{\prime }-y^{\prime }$$

Since we have a single transformation matrix that can be applied to the whole image, the computation speed is notably higher in comparison to a piecewise affine transformation that we will delve into in Section 4.2.

However, compared to the piecewise affine transformation proposed in the next section, two main disadvantages of a projective transformation to note are that it will modify the curvature of the viewing window, as well as the distance between horizontal lines. This can be seen in Figure 4. We can observe that the lower sections became significantly more elongated. This means that the alteration in slope must remain minor to ensure the resultant output remains realistic.

Projective transforms also fail when the top two points, $p1_{\mathrm{left}}$ and $p1_{right}$, are close together. This is because a projective transform is really treating the transformation as a transformation from 2D projection coordinates to 3D real world coordinates, then back to a different 2D projection. As such, the two top points being close together gets treated as a vanishing point on the horizon, which would be extremely far away in real world coordinates. This large value then distorts the transformation when we go back into the second 2D projection coordinates.

During experiments, when generating a random transform, the new slope was sampled from a normal distribution with scale 0.25, centered on the original slope value. The new slope was further limited to be in the range [0.75, 100] to ensure that the deviation from the original image is not too extreme. Linear images transformed using this method were sampled with a mean slope of 10. The minimum distance between $p1_{\mathrm{left}}$ and $p1_{right}$ was set to 10 to limit extreme distortion

### 4.2. Piecewise Affine Transform

An alternative approach to employing a projective transformation would be utilizing a piecewise affine transformation. In this case, rather than a single transformation matrix that would be applied to the entire image, a number of affine transformations are estimated for different sections of the image based on a grid of points. The grid of points is generated by first locating the virtual focal point, after which the grid is established using polar coordinates originating from that center point as a distance and angle. This implementation assumes circular radiation from the center. This grid is generated for each new and old group of corner points. Visualization of such a grid can be seen in Figure 5.

The virtual focal point is calculated finding the point which intersects with the two lines defined by the left and right bounds of the angled viewing window. A random transformation is sampled similarly to the projective transform through finding new corner points, as shown in Figure 3.

This transformation requires significantly more computing time compared to using projective transforms, as it requires estimating the transformation for multiple sections of the image. An insufficient density of points also results in artifacts between each section of the image. With a sufficient density of points, however, the results of this transformation look significantly more realistic compared to the simpler projective transform and can handle a greater change in slope with few distortions. Examples of both transforms can be seen in Figure 2.

During experiments, when generating a random transform, the new slope was sampled from a normal distribution with scale 0.25, centered on the original slope value. The new slope was further limited to be in the range [0.75, 100] to ensure that the deviation from the original image is not too extreme. Linear images transformed using this method were sampled with a mean slope of 10. The point grid contained eight columns and four rows to balance compute requirements and resolution.

### 4.3. COVID-Net L2C-ULTRA Architecture Design

To explore the impact of extended linear–convex ultrasound augmentation learning in a clinical resource-constrained scenario, we introduce COVID-Net L2C-ULTRA, a convolutional neural network for COVID-19 assessment and screening using POCUS that balances efficiency and representation capacity. The architectural design comprises a deep columnar anti-aliased residual architecture with highly customized macro- and micro-architecture designs that were automatically determined through a machine-driven design exploration strategy, called generative synthesis [36], for high computational and representational efficiency. Generative synthesis is a machine learning strategy that uses a generator function to create a variety of unique deep neural networks based on a given seed input. This generator works in tandem with an inquisitor, which learns to improve the neural networks by observing their responses to targeted stimuli and guiding updates to the generator’s parameters. The result is an ensemble of highly efficient neural networks that satisfy operational requirements, demonstrating the versatility and efficacy of generative synthesis in deep learning design. The architecture design used in this experiment has 636 M FLOPs and 4.69 M parameters.

The advantage of ultrasound imaging systems lies in their cost-effectiveness and energy efficiency. As such, using generative synthesis allows us to maintain that power and computational efficiency by automatically striking a balance between detection accuracy and network performance. The COVID-Net L2C-ULTRA network architecture design features progressively increasing inter-column connectivity as we go deeper into the network. Such variations in inter-column connectivity enable greater efficiency in the lower degrees of abstraction while achieving greater representational complexity at higher degrees of abstraction. Furthermore, the COVID-Net L2C-ULTRA network architecture exhibits anti-aliasing architectural characteristics, which enables greater representational stability. This, in turn, allows the network to achieve enhanced robustness and generalization [37], a particularly crucial factor for dependable clinical decision support applications.

COVID-Net L2C-ULTRA’s explainability is assessed using the GSInquire approach [38]. GSInquire identifies essential elements within an input image that significantly influence the network’s decisions through a generative synthesis approach. The result is an annotated image in which the critical regions are highlighted, potentially leading to a significant alteration in the classification outcome if removed.

## 5. Results

To evaluate the impact of the proposed extended linear-convex ultrasound augmentation learning strategy, we used the COVIDx-US v1.4 benchmark data set [33] in this study. As explained in Section 3, the dataset consists of 16,649 POCUS images, with 12,260 captured using convex probes and 4389 captured using linear probes, categorized into 6764 COVID-19-positive and 9985 COVID-19-negative images. This dataset is larger and more diverse than the dataset leveraged in [5]. The train, validation, and testing splits are 72%, 14%, and 14%, respectively. In this section, we present the results of the experiments as well as the clinical validation.

### 5.1. Assessing the Impact of Including Linear Probe Images

We evaluate the effectiveness of including linear probe images while training with and without the proposed method. Furthermore, one argument may be that the proposed piecewise affine transform is a data augmentation method, which improves the result of the model by introducing further diversity to the dataset. As such, we further evaluate the viability of the proposed piecewise affine transform against a number of common data augmentation methods, including rotation, cropping, horizontal flips, and color (brightness) augmentations. Results can be seen in Table 1:

Table 1 compares two objectives: the ability to effectively utilize linear probe images to improve images (seen in the difference in metric performance between with and without linear images), as well as the general ability of each augmentation method to improve performance (seen in the overall metric score of each method).

Only the proposed piecewise affine transformation method demonstrated a consistent ability to effectively utilize the inclusion of linear probe images to improve performance in all metrics. In particular, all other methods exhibited a decrease in recall performance after the inclusion of linear images, while the proposed method was able to improve this by ≈5%, raising recall from 77.9% to 83%. In other metrics such as accuracy, AUC, and precision, the proposed method maintained a solid improvement in performance through the inclusion of linear images in the dataset, showing an effective utilization of the new training data. The proposed method also consistently showed the overall highest performance in all metrics except precision, where it performed second best, proving its reliability in the second objective of generally improving model performance through data augmentation. Compared to the case of no data augmentation, the proposed method with linear images shows a gain of 3.9% in test accuracy, 3.2% in AUC, 10.9% in recall, and 4.4% in precision.

Table 2 compares the proposed approach against transforming the convex images through linear rectification. This approach, seen in Yaron et al. [30] takes the polar coordinates of convex probe images and transforms it into Cartesian coordinates. Different from the data augmentation approaches compared in Table 1, this existing approach is explicitly to aid in decreasing the heterogeneity between different probes. An example of a transformed image can be seen in Figure 6. The polar coordinates are sampled the same way as in the piecewise affine transform.

The proposed approach (piecewise affine) still offers better performance in nearly all metrics, and demonstrates more effective utilization of linear probe images after its inclusion in the dataset.

### 5.2. Clinical Validation and Network Explainability Analysis

As explained in Section 4, to confirm the viability of the COVID-Net L2C-ULTRA prediction results, we apply GSInquire [38] to the ultrasound image inputs. Our contributing clinician (A.F.), who possesses expertise in ultrasound image analysis, randomly reviewed the outcomes of GSInquire annotation and reported his findings and observations. This assessment aimed to determine whether the network effectively and reliably captures clinically important patterns. Our contributing clinician (A.F.) is an Assistant Professor in the Department of Emergency Medicine, and serves as the ultrasound co-director for undergraduate medical students at McGill University. He is practicing Emergency Medicine full-time at Saint Mary’s Hospital in Montreal. He was provided with the results of the proposed model on test set to confirm its validity. Of the images reviewed, four examples, shown in Figure 7, were chosen as examples to demonstrate the model’s performance on COVID-19, pneumonia, and normal annotated images. The summary of our expert clinician’s report is as follows.

**Case 1 (Figure 7a).** This image demonstrates the presence of b-lines throughout the scanned window which is assumed to be an intercostal space. The pleura appears to be broken and irregular and there is a suspicion of subpleural consolidations. In the appropriate clinical context, this appearance would be one of the classic lung ultrasound presentations of COVID-19.

**Case 2 (Figure 7b).** In this image, it is not very clear what GSINquire is focusing on and another frame from this clip would be needed to make a better interpretation. Nevertheless, the pleural and subpleural region emerges as the pivotal element. Considering the peripheral location of COVID-19 lesions within the lungs, this area precisely aligns with our anticipation of locating pathological indicators.

**Case 3 (Figure 7c).** This image denotes lung hepatization and a classic appearance of consolidated lung, which is a feature of pneumonia.

**Case 4 (Figure 7d).** This image denotes the classic appearance of the lung on ultrasound with the presence of the pleura and the artifacts called a-lines, which are present in normal lungs.

The observations and findings from our expert clinician substantiated the presence of pertinent artifacts and markers associated with lung diseases, such as consolidations and b-lines, along with indicators of healthy lungs, confirming the reliability of COVID-Net L2C-ULTRA in capturing clinically relevant features.

## 6. Discussion and Conclusions

In the fight against the COVID-19 pandemic, researchers and technologists have turned to innovative solutions that leverage cutting-edge technologies. Among these, the integration of ultrasound imaging into AI-powered screening and detection methods has emerged as a promising avenue. Unlike other modalities, ultrasound offers distinct advantages in the context of COVID-19 assessment. This imaging technique provides non-invasive insights into the lung’s structural dynamics, making it particularly suitable for identifying abnormalities associated with the disease. Additionally, the portability, safety, and cost-effectiveness of ultrasound equipment position it as an accessible and valuable tool, especially in resource-limited settings. In this rapidly evolving landscape, harnessing the potential of ultrasound images coupled with AI holds great promise for enhancing the accuracy and efficiency of COVID-19 screening and detection protocols.

In this research, we presented an extended linear–convex ultrasound augmentation learning strategy known as COVID-Net L2C-ULTRA, designed to optimize deep learning networks for COVID-19 detection and screening via POCUS images. By utilizing a diverse benchmark dataset, COVIDx-US, we demonstrated that our method exhibits superior performance when linear images are integrated. The method also showcased overall high performance across multiple metrics, underlining its capacity to proficiently handle various ultrasound image types.

A significant result from our experiments was the remarkable improvement in recall performance following the inclusion of linear images through the proposed piecewise affine transform, whereas alternative methods showed a decrease in recall performance. This finding is crucial, as maintaining high recall rates is of utmost importance in clinical decision support applications, mitigating the risk of missing critical diagnoses or medical conditions by minimizing false negatives.

The proposed approach also had more effective utilization of linear images, as well as overall higher performance, compared to performing linear rectification on the convex images. This is likely due to two factors.

Firstly, the experimental dataset consisted primarily of convex images with limited linear images. Linear rectification transforms the convex images, meaning most images in the dataset needed to be majorly modified, which may introduce artifacts. In contract, the proposed method only subtly transforms the convex images, while only linear images goes through more extreme modification. As such, the results may differ significantly if the model was trained and evaluated using a dataset with a higher distribution of linear images.

Second, the proposed approach doubles as a data augmentation approach, while linear rectification only transforms the convex images to be more similar to linear images. As such, the proposed approach allows for greater diversity during training that linear rectification cannot offer.

The GSInquire tool was employed for explainability analysis, providing further validation for our strategy. It confirmed that our network focuses on the relevant regions in the ultrasound images for COVID-19 detection. This transparency is essential in the creation of AI-based medical tools, as it uncovers the decision-making process, contributing to trust-building among healthcare practitioners.

It is important to note that while data augmentation is a valuable technique, it cannot completely replace the need for a diverse and representative dataset. It is still essential to collaborate with medical institutions to collect as much relevant and accurate data as possible. Data augmentation should be used as a complementary strategy to enhance training when the original dataset is limited in size. Moving forward, our team is currently engaged in developing a protocol for an upcoming study, aiming to uniformly capture ultrasound images and videos from a predefined group of consented participants. Moreover, the piecewise affine transform introduces significant computational cost during training (and inference, in the case of linear images), especially in comparison to the projective transform. Future work may look into more computationally efficient methods.

In addition, linear and convex probes differ in other aspects not covered by the proposed transformation, such as tissue depth. As such, transformed linear images cannot be taken as visually identical to a corresponding convex images. The goal of the proposed approach is to decrease the visual differences enough to allow for greater utilization of linear images during training of machine vision models.

Our study concludes that the extended linear–convex ultrasound augmentation learning strategy demonstrates effectiveness in enhancing COVID-19 detection in POCUS images using the COVID-Net L2C-ULTRA deep neural network. The incorporation of this augmentation technique holds promise for the development of more accurate and reliable AI-based clinical decision support systems for COVID-19 detection.

The successful enhancement of the model’s performance using our method underscores the potential role of AI-based tools in clinical settings. They can significantly assist healthcare professionals in diagnosing and managing COVID-19 patients. As we continually refine and build upon the techniques presented in this study, we anticipate further advancements in AI-driven medical imaging analysis, leading to improved patient outcomes.

Looking toward the future, potential avenues for research could include the exploration of data augmentation techniques specifically designed for POCUS images and the investigation of how the proposed method can be modified for other medical imaging modalities or conditions. Additionally, integrating COVID-Net L2C-ULTRA with additional clinical information and patient data may lead to the development of a more comprehensive and robust decision support system for managing COVID-19 patients.

## Figures and Tables

**Figure 1 sensors-24-01664-f001:**
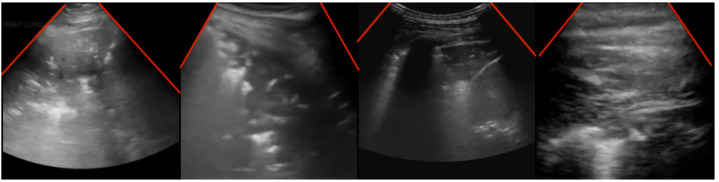
An example of ultrasound images with different viewing windows. The bounds of the viewing windows are marked in red.

**Figure 2 sensors-24-01664-f002:**
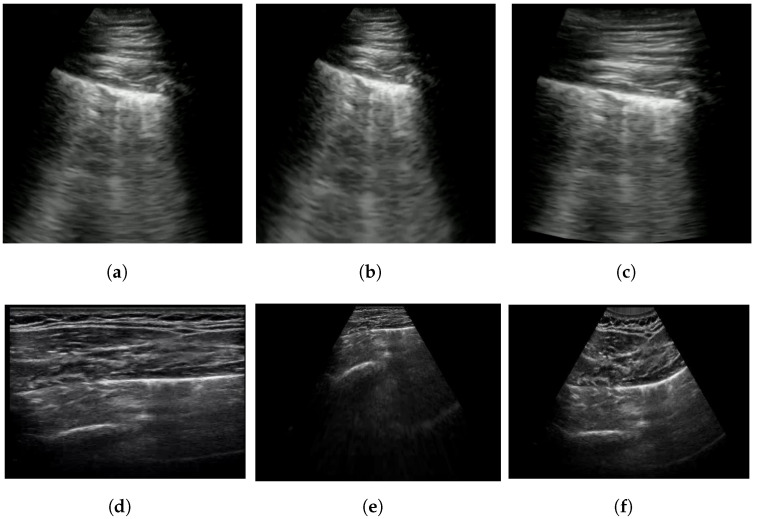
Proposed image transformations on convex ultrasound images (**b**,**c**) and linear ultrasound images (**e**,**f**). (**b**,**e**) is generated using projective transform, while (**c**,**e**) is generated using piecewise affine transform. The original images prior to transformations are shown in (**a**,**d**).

**Figure 3 sensors-24-01664-f003:**
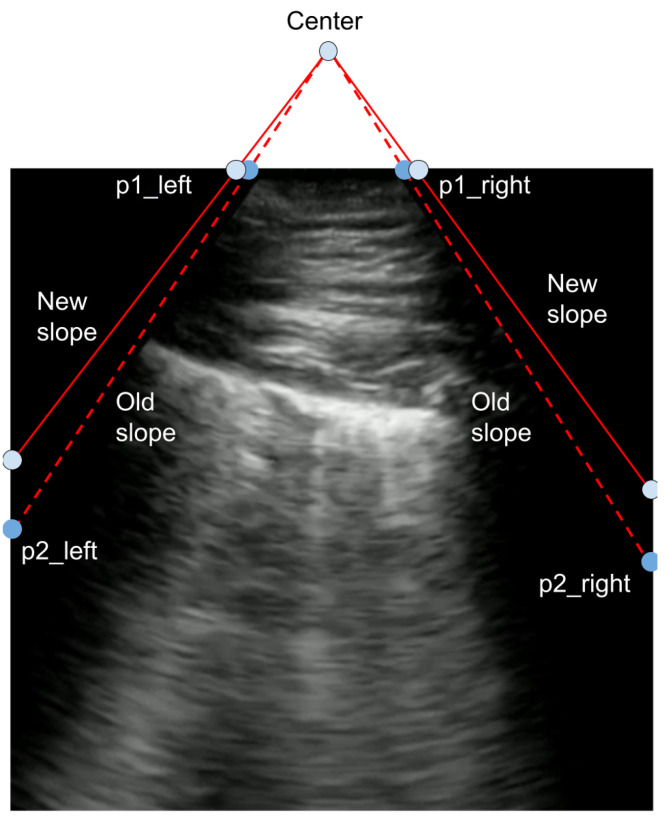
To generate a random transformation: 1. Using the labeled bounds of a convex ultrasound image defined by the corner points $\left\{p1_{\mathrm{left}},p2_{\mathrm{left}},p1_{\mathrm{right}},p2_{\mathrm{right}}\right\}$, find a new slope based on the distribution N(old_slope,σ). 2. Define new points $\left\{p1_{\mathrm{left}},p1_{\mathrm{right}}\right\}$ using the new slope and constant center point. 3. Estimate a transformation matrix based on the new and old set of corner points.

**Figure 4 sensors-24-01664-f004:**
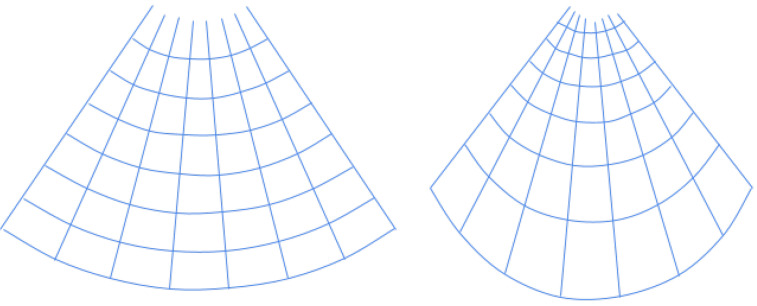
How a projective transform will modify a convex ultrasound image.

**Figure 5 sensors-24-01664-f005:**
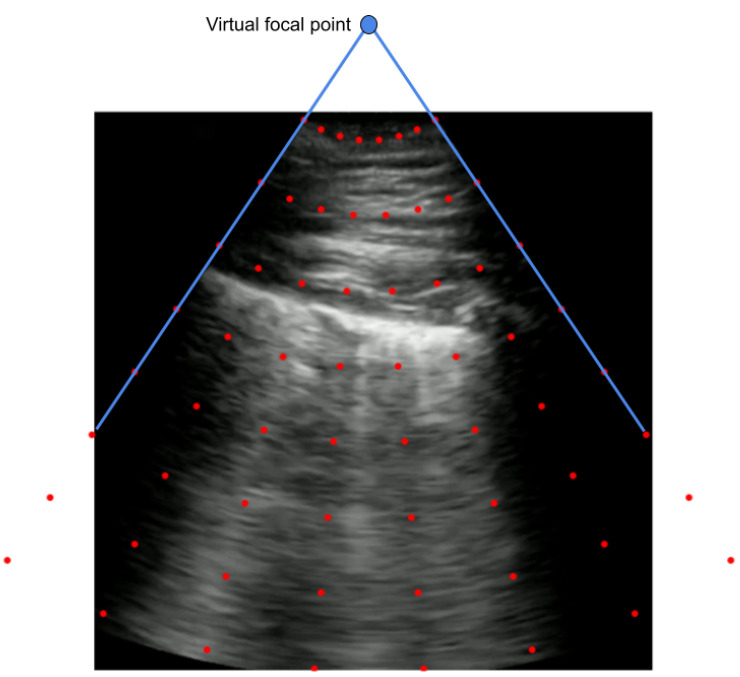
An example grid of points for constructing a piecewise affine transform.

**Figure 6 sensors-24-01664-f006:**
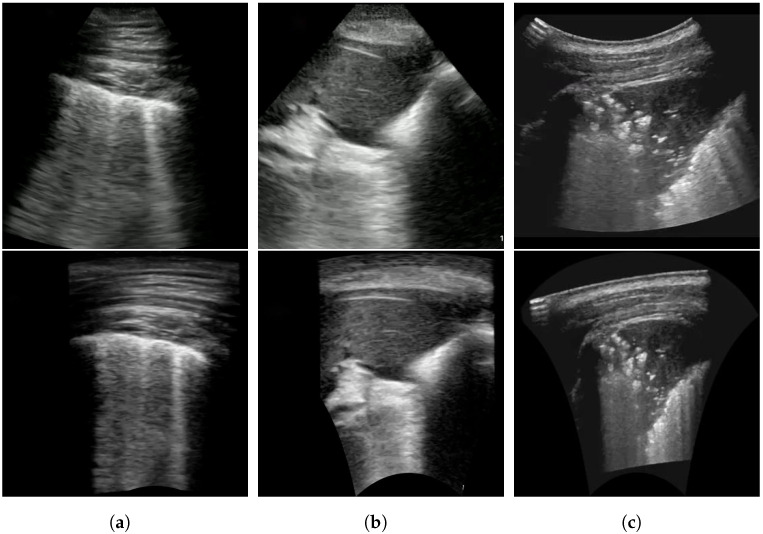
Sample ultrasound images, after linear rectification, as seen in Yaron et al. [30]. The original image is shown on top while the transformed image is shown on the bottom. An ideal example can be seen in (**a**). An example where the original viewing window is cut off to be flat on the bottom, causing the resultant transformed image to have a curved black outline on the bottom is shown in (**b**). Finally, (**c**) shows an example of how poorly labeled viewing window corners can result in a rotated final image.

**Figure 7 sensors-24-01664-f007:**
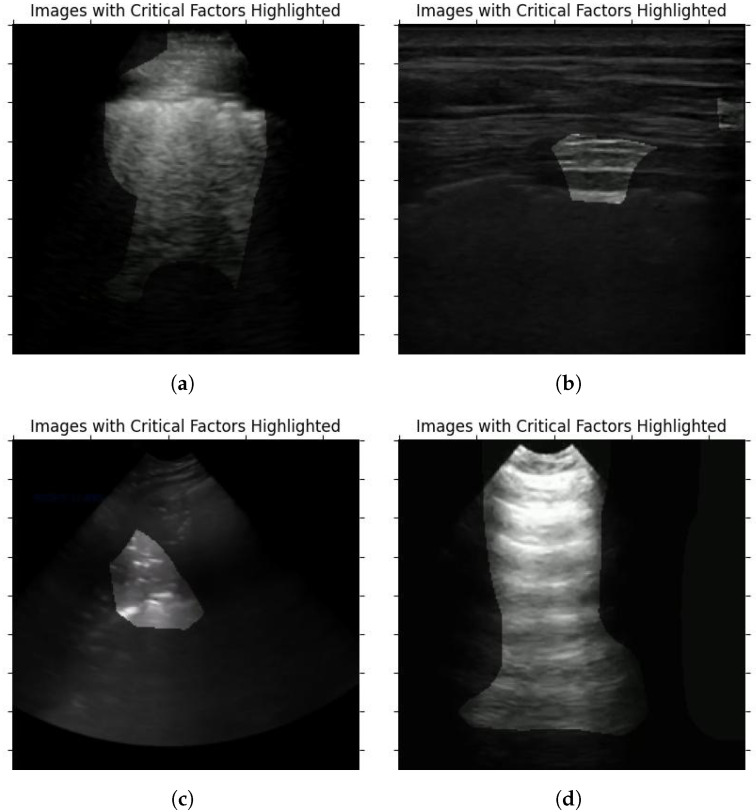
Sample ultrasound images, annotated by GSInquire, reviewed and reported on by our contributing clinician. (**a**) COVID-19 example; (**b**) COVID-19 example; (**c**) pneumonia example; (**d**) normal example.

**Table 1 sensors-24-01664-t001:** Comparison of different augmentation methods before and after the inclusion of linear probe images. The best-performing score in each row is highlighted in bold. The color serves as additional cue to differenciate between a decrease (red) compared to an increase (green).

		Piecewise Affine	Projective	Rotation	Crop	Flip	Color	None
Accuracy	w/o linear	**88.6**	88.1	84	86	86.3	85.5	86.2
	w/ linear	**90.6**	87.4	86.8	86.3	85.8	85.6	86.7
	difference	↑2.0%	↓0.7%	↑2.8%	↑0.3%	↓0.5%	↑0.1%	↑0.5%
AUC	w/o linear	**95**	94.1	92.3	93.5	93	92	93.6
	w/ linear	**97.1**	94.5	94.5	94	94	89.9	93.9
	difference	↑2.1%	↑0.4%	↑2.2%	↑0.5%	↑1.0%	↓2.1%	↑0.3%
Recall	w/o linear	77.9	73	79	75.8	**79.2**	76.6	78.4
	w/ linear	**83**	70.6	74.8	73.7	71.1	69.1	72.1
	difference	↑5.1%	↓2.4%	↓4.2%	↓2.1%	↓8.1%	↓7.5%	↓6.3%
Precision	w/o linear	86.4	**88.9**	76.5	81.2	80.5	77.4	81.3
	w/ linear	88.8	**90.8**	84.2	84.1	83.4	81.4	84.4
	difference	↑2.4%	↑1.9%	↑7.7%	↑2.9%	↑2.9%	↑4.0%	↑3.1%

**Table 2 sensors-24-01664-t002:** Comparison against linear rectification method before and after the inclusion of linear probe images. The best performing score in each row is highlighted in bold. The color serves as additional cue to differenciate between a decrease (red) compared to an increase (green).

		Piecewise Affine	Projective	Linear Rectification	None
Accuracy	w/o linear	**88.6**	88.1	87.2	86.2
	w/ linear	**90.6**	87.4	86.5	86.7
	difference	↑2.0%	↓0.7%	↓0.7%	↑0.5%
AUC	w/o linear	**95**	94.1	94.8	93.6
	w/ linear	**97.1**	94.5	94.9	93.9
	difference	↑2.1%	↑0.4%	↑0.1%	↑0.3%
Recall	w/o linear	77.9	73	**87.3**	78.4
	w/ linear	**83**	70.6	78.7	72.1
	difference	↑5.1%	↓2.4%	↓8.6%	↓6.3%
Precision	w/o linear	86.4	**88.9**	84.8	81.3
	w/ linear	88.8	**90.8**	80.2	84.4
	difference	↑2.4%	↑1.9%	↓4.2%	↑3.1%

## Data Availability

The dataset used in the study is the COVIDx-US (link: https://github.com/nrc-cnrc/COVID-US, accessed on 2 January 2024) which is a public benchmark dataset of COVID-19-related ultrasound images and videos, collected, curated and released by our team.

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
