# Peer review of "COVID-Net L2C-ULTRA: An Explainable Linear-Convex Ultrasound Augmentation Learning Framework to Improve COVID-19 Assessment and Monitoring"

_sensors, 2024, doi:10.3390/s24051664_

Round 1

Reviewer 1 Report

Comments and Suggestions for Authors

The paper needs major revision. 

Concern 1. Lack of Detailed Explanation of the Proposed Approach: The paper does not provide a sufficiently detailed explanation of the proposed framework, particularly the technical specifics of the projective and piecewise affine transformations used in the data augmentation process. More in-depth details on the methodology would be beneficial for reproducibility and understanding.

Concern 2. Insufficient Validation and Testing: The clinical validation seems limited. There is a need for more comprehensive testing, ideally involving multiple clinicians and a larger dataset, to establish the robustness and generalizability of the approach.

Concern 3. Limited Discussion on Limitations and Challenges: The paper does not adequately discuss the limitations and potential challenges of the proposed approach. Including a section on limitations would provide a more balanced view and help future researchers in addressing these issues.

Concern 4. Comparison with Existing Methods: There is a lack of comparison with existing methods or baseline models. This comparison is essential to demonstrate the effectiveness and advancement of the proposed framework over current techniques.

Addressing these concerns would significantly strengthen the paper, enhancing its scientific rigor, clarity, and potential for practical application in clinical settings.

Reviewer 2 Report

Comments and Suggestions for Authors

1. Clarification on Dataset Composition:

It is crucial for the manuscript to explicitly mention the number of unique patients from whom the COVIDx-US v1.4 dataset was derived. While the authors specify a 72:14:14 ratio for dividing the dataset into training, validation, and test sets, it remains unclear if, for instance, two ultrasound images from the same patient could be assigned to different sets (e.g., one to the training set and the other to the test set). Providing information on the unique count of COVID and non-COVID patients in the dataset, along with the approximate number of images obtained from each, is essential. Understanding the distribution of CT samples taken from patients in both groups is critical to assessing any potential bias and, consequently, instilling confidence in the study's outcomes.

 2. Heterogeneity in Imaging and Depth Considerations:

The distinction between linear and convex probes is essential for understanding the imaging variations in the dataset. Linear probes are commonly used by doctors when they aim to visualize tissues close to the surface, whereas convex probes are preferred for examining deeper tissues. While my expertise doesn't specifically cover details related to probes, it's worth noting that my knowledge is not exhaustive in this specialized area. Nonetheless, the authors have successfully addressed dataset heterogeneity through geometric transformations. However, it is imperative to consider whether the potential variations in image depths, given the differences between linear and convex probes, have been adequately controlled for. In essence, have the authors accounted for heterogeneity arising from the diverse tissue depths captured by these probes during their analysis? Clarifying this point will contribute to a more comprehensive evaluation of the study's findings.

Minor Note: In Table 1, increases and decreases are highlighted using green and red fonts. However, this color-coding may cause discomfort for readers with color vision deficiencies. It is suggested to use black font for better readability. Nevertheless, the authors may exercise their discretion in this matter.

Round 2

Reviewer 1 Report

Comments and Suggestions for Authors

This research paper presents an innovative framework for enhancing COVID-19 assessment through advanced ultrasound image processing. The authors address a significant challenge in the clinical workflow of COVID-19 diagnosis by leveraging artificial intelligence to augment ultrasound imaging data from both convex and linear probes, thus improving the model's generalization capabilities. Their methodological approach, incorporating projective and piecewise affine transformations, aims to overcome the heterogeneity in ultrasound image appearances, enhancing the performance of deep neural networks for COVID-19 assessment.

The paper reports a noticeable improvement in various performance metrics, including test accuracy, AUC, recall, and precision, when applying their extended linear-convex ultrasound augmentation learning. Specifically, the inclusion of linear probe images, transformed to resemble convex probe data, resulted in a marked performance boost, underscoring the method's effectiveness in utilizing available data more efficiently.

Despite the authors' acknowledgment of the study's limitations and areas for future research, the contribution, while not groundbreaking, adds value to the field of medical imaging and AI-driven diagnostics. The proposed framework demonstrates potential for enhancing the accuracy and reliability of COVID-19 screening tools, particularly in resource-constrained settings where the availability of expert clinicians is limited.

In conclusion, while the paper may not represent a significant leap forward in the field, its contribution to improving COVID-19 assessment through innovative ultrasound image augmentation and the use of explainable AI techniques warrants acceptance. The findings offer a promising avenue for further research and development in the application of AI in medical imaging, potentially leading to more effective and accessible diagnostic solutions for COVID-19 and beyond.

Reviewer 2 Report

Comments and Suggestions for Authors

I believe that the changes made by the authors in response to my suggestions are sufficient.